# *VL Norm*: Rethink Loss Aggregation in RLVR

## Abstract

We propose *VL Norm* (**V**ariance-reduced **L**ength-dependent **Norm**alization), a simple yet effective loss aggregation method tailored to the characteristic of *dynamic generation lengths* in Reinforcement Learning with Verifiable Rewards (RLVR). Recently, RLVR has demonstrated strong potential in improving the reasoning capabilities of large language models (LLMs), but a major challenge lies in the large variability of response lengths during training, which leads to high gradient variance and unstable optimization. Although previous methods such as GRPO, DAPO, and Dr. GRPO introduce different loss normalization terms to address this issue, they either produce biased estimates or still suffer from high gradient variance. By analyzing the effect of varying lengths on policy loss both theoretically and empirically, we reformulate the problem as finding a minimum-variance unbiased estimator. Our proposed *VL Norm* not only provides an unbiased estimate of the true policy loss but also minimizes gradient variance in theory. Extensive experiments show that it consistently achieves superior results across different model sizes, maximum lengths, and tasks.

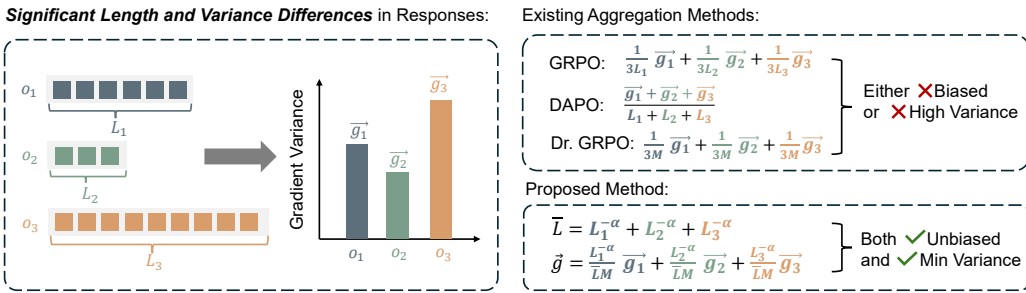

Figure 1: **Left:** In RLVR, trajectory lengths vary significantly, and long trajectories induce high gradient variance, causing unstable training. **Right:** Existing gradient aggregation methods across different lengths either lead to biased updates or suffer from high variance. In this paper, we propose a new aggregation method, *VL Norm*, that is both unbiased and variance-minimized.

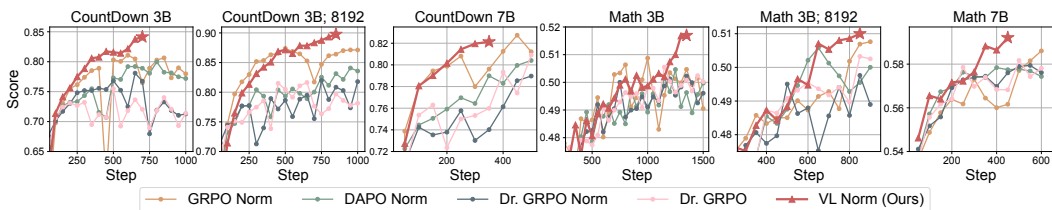

Figure 2: Training dynamics of *VL Norm* compared with baselines across tasks (CountDown, Math), maximum lengths (3072, 8192), and model sizes (3B, 7B). Performance is measured by Avg@8 on CountDown and by a weighted Avg@8 across four math datasets. *VL Norm* consistently yields more stable training and consistently converges to higher accuracy.

# 1 INTRODUCTION

Recently, significant progress has been made in improving the reasoning abilities of Large Language Models (LLMs) (Guo et al., 2025; Jaech et al., 2024; Team et al., 2025)) The key of these advances is Reinforcement Learning with Verifiable Rewards (RLVR), which applies rule-based rewards and uses reinforcement learning to update model weights accordingly (Shao et al., 2024; Liu et al., 2025; Yu et al., 2025). Unlike traditional reinforcement learning, RLVR introduces a unique challenge: the response trajectories vary drastically in length, ranging from a few dozen to several thousand tokens, and often growing longer as training progresses (Team et al., 2025; Yu et al., 2025). Such variability complicates the optimization process. Prior studies have shown that rapid fluctuations in response length can lead to model accuracy collapse (Dai et al., 2025), truncated overlong responses may trigger entropy explosion and degrade performance (Yu et al., 2025), and excessively long responses introduce high variance and destabilize training (Zheng et al., 2025).

To address these issues, the common practice is applying *length-dependent* normalization in loss aggregation. Specifically, GRPO applies a sample-level normalization, dividing each sample-level loss by its response length (Shao et al., 2024); while DAPO uses a batch-level approach, normalizing the total loss by the sum of response lengths in the batch (Yu et al., 2025). Intuitively, these length-dependent aggregation methods normalize longer responses or batches, which stabilizes training by preventing long responses from dominating. In addition, because such length-dependent factors deviate from standard reinforcement learning theory, Dr. GRPO, in contrast, avoids any length-dependent factor and normalizes the gradient with a fixed constant (Liu et al., 2025).

While these aggregation strategies have shown empirical benefits, their theoretical properties remain largely underexplored. There is a lack of analysis on how they influence the statistical properties of gradient estimation in RLVR, with gradient variance being particularly important because high variance leads to inefficient training and even model collapse (Williams, 1992; Sutton et al., 1998; Schulman et al., 2015). We illustrate this comparison in Figure 4. Estimators with higher gradient variance are more prone to deviating from the optimal optimization direction, which hinders effective training.

In this work, we present the first systematic analysis of how different aggregation strategies affect gradient estimation in RLVR training. We begin by reformulating the aggregated gradient as a linear combination of sample-level unnormalized gradients. Then we observe directly that the variance of each unnormalized gradient grows linearly with its response length. With this perspective in place, we carefully examine the bias–variance properties of existing methods and uncover two major issues: (1) The aggregation strategies in GRPO and DAPO introduce length-dependent bias in estimating the true policy gradient. As response lengths increase, their parameter updates shrink in gradient norm, slowing convergence. (2) The aggregation strategies in DAPO and Dr. GRPO lead to a higher coefficient of variation (CV). Since CV measures normalized variance, a higher value means greater relative noise for the same gradient norm, resulting in less stable training.

Motivated by these observations, we propose **V**ariance-reduced **L**ength-dependent **Norm**alization (*VL Norm* for short), an unbiased loss aggregation technique with minimized variance. These two properties make it a stable update algorithm and enable convergence to better model accuracy. In addition, *VL Norm* is simple to implement and requires fewer than ten lines of code changes.

We conduct extensive experiments on two models, Qwen2.5-3B and Qwen2.5-7B (Yang et al., 2024), across two tasks: CountDown and Math, and across different maximum response lengths. Results show that the proposed *VL Norm* not only stabilizes the training but also achieves higher accuracy. We further compare our approach with alternative strategies for handling long responses, including Overlong filtering and Soft punishment from DAPO (Yu et al., 2025), and find that *VL Norm* achieves significant performance gains over these methods.

# 2 ANALYSIS OF LOSS AGGREGATION METHODS IN RLVR

## 2.1 REFORMULATE EXISTING METHODS FROM A UNIFIED PERSPECTIVE

Given a parameterized policy $\pi_\theta$ and trajectories $\tau = \{(s_t, a_t, r_t)\}_{t=0}^T$ sampled from $\pi_\theta$ and the environment, vanilla policy gradient updates $\pi_\theta$ by the following gradient estimator:

$$\nabla_\theta J(\theta) = \mathbb{E}_{\tau \sim \pi_\theta} \left[ \sum_{t=0}^{T} A_t \nabla_\theta \log \pi_\theta(a_t \mid s_t) \right] \tag{1}$$

where $J(\theta)$ is the expected return and $A_t$ is advantage function (Williams, 1992). Recently, reinforcement Learning with Verifiable Reward (RLVR) has achieved notable success in boosting reasoning capabilities of large language models (LLMs). A representative method is Group Relative Policy Optimization (GRPO), which eliminates the need for a critic model by instead estimating the baseline within each group (Shao et al., 2024). For a given question $q$, GRPO first sample a group of outputs $o_1, o_2, ..., o_G$ from the old policy $\pi_{\theta_{old}}$, and update current $\pi_\theta$ by:

$$\nabla_\theta \frac{1}{G} \sum_{i=1}^{G} \frac{1}{L_i} \sum_{t=1}^{L_i} \min \left( r_{i,t}(\theta) \, A_i, \ \mathrm{clip} \left( r_{i,t}(\theta), \, 1 - \epsilon, \, 1 + \epsilon \right) A_i \right),$$

where $L_i = |o_i|$ denotes the length of the $i$-th response, and $r_{i,t}(\theta) = \frac{\pi_\theta(o_{i,t}|q,o_{i,<t})}{\pi_{\theta_{old}}(o_{i,t}|q,o_{i,<t})}$ is the token-level importance sampling ratio. The advantage $A_i$ is computed via group-level normalization $A_i = \frac{r_i - \mathrm{mean}(\{r_1, r_2, ..., r_G\})}{\mathrm{std}(\{r_1, r_2, ..., r_G\})}$. Notably, GRPO first normalizes the per-sample gradient by response length $L_i$, and then averages across the group, which has been observed to have an impact on the algorithm performance in prior works (Yu et al., 2025). To analyze the underlying effect, we define the unnormalized sample-level gradient $\boldsymbol{g}_i$ as:

$$\boldsymbol{g}_i = \nabla_\theta \sum_{t=1}^{L_i} \min \left( r_{i,t}(\theta) \, A_i, \ \mathrm{clip} \left( r_{i,t}(\theta), \, 1 - \epsilon, \, 1 + \epsilon \right) A_i \right).$$

If we omit clipping and importance sampling, the unnormalized gradient $\boldsymbol{g}_i$ approximates the standard policy gradient estimator with advantage estimation:

$$\boldsymbol{g}_i \approx \sum_{t=1}^{L_i} A_i \nabla_\theta \log \pi_\theta(o_{i,t} \mid q, o_{i,<t}),$$

This aligns with Equation 1, thus the expectation of $\boldsymbol{g}_i$ also aligns with the true policy gradient, which translates to:

$$\mathbb{E}_{o_i \sim \pi_\theta}[\boldsymbol{g}_i] \approx \nabla_\theta J(\theta).$$

Although $\mathbb{E}[\boldsymbol{g}_i] \approx \nabla_\theta J(\theta)$ holds, GRPO does not update the model using the maximum likelihood estimator $\frac{1}{G} \sum_{i=1}^{G} \boldsymbol{g}_i$. Instead, it applies an additional length-based normalization, estimating the gradient as $\frac{1}{G} \sum_{i=1}^{G} \frac{1}{L_i} \boldsymbol{g}_i$. Beyond GRPO, alternative loss aggregation strategies have been proposed by DAPO and Dr. GRPO, given by:

$$\boldsymbol{g}_{\mathrm{DAPO}} = \frac{1}{\sum_{i=1}^{G} L_i} \sum_{i=1}^{G} \boldsymbol{g_i} \, ; \qquad \boldsymbol{g}_{\mathrm{Dr.GRPO}} = \frac{1}{GM} \sum_{i=1}^{G} \boldsymbol{g_i}$$

DAPO normalizes the gradient by the total response length within a batch (Yu et al., 2025), whereas Dr. GRPO applies a normalization using a constant $M$ followed by sample-level normalization (Liu et al., 2025). Since GRPO and DAPO both rely directly on response lengths, we refer to their aggregation methods as *length-dependent*. Dr. GRPO, however, is *length-independent*, as its normalization does not vary with response length during training.

We acknowledge that the computation of each $\boldsymbol{g}_i$ may differ across methods: for instance, DAPO introduces additional techniques such as dynamic sampling, varied clipping ratios, and overlong filtering or punishment; Dr. GRPO, in contrast, estimates the advantage using $A_i = r_i - $

mean$(r_1, \ldots, r_G)$ rather than the original form $A_i = \frac{r_i - \text{mean}(r_1, \ldots, r_G)}{\text{std}(r_1, \ldots, r_G)}$. These auxiliary modifications, however, are orthogonal to the loss aggregation methods. For simplicity, our analysis focuses on aggregation methods, and we will provide further empirical studies on these auxiliary modifications in the evaluation section.

## 2.2 GRADIENT VARIANCE GROWS PROPORTIONALLY WITH RESPONSE LENGTH

Both GRPO and DAPO adopt length-dependent aggregation methods. We hypothesize that their designs are motivated by the statistical property that *the variance of the unnormalized gradient $g_i$ grows approximately proportionally to the response length*.

Consider $\text{Var}(g_i) \approx \text{Var}(\sum_{t=1}^{L_i} A_i \nabla_\theta \log \pi_\theta(o_{i,t} \mid q, o_{i,<t}))$ [1]. We approximate this as $\text{Var}(g_i) \approx \sum_{t=1}^{L_i} \text{Var}(A_i \nabla_\theta \log \pi_\theta(o_{i,t} \mid q, o_{i,<t}))$, by ignoring covariance between individual token-level gradients. Assuming each token-level term contributes approximately a constant variance $V$, we thus have $\text{Var}(g_i) \approx V \cdot L_i$. This approximation indicates that samples with longer responses inherently exhibit higher gradient variance due to increased randomness, and the gradient variance grows proportionally with response length.

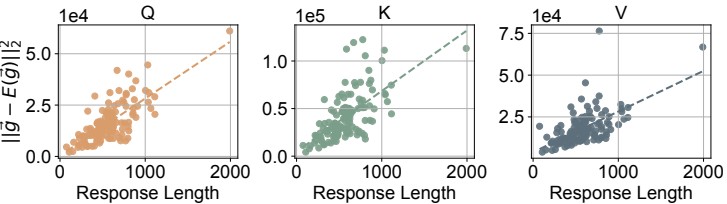 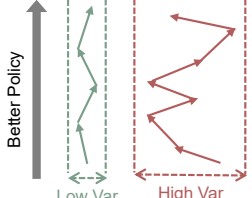

Figure 3: Deviation $||g_i - \mathbb{E}[g_i]||^2$ for a random selected sample on the Q, K, V projection in the last layer. $\mathbb{E}[g_i]$ is estimated by the average of 128 rollouts.

Figure 4: Comparison between low and high gradient variance.

We also verify this property empirically using the Qwen2.5-7B model. We randomly select one math question from the Open Reasoner Zero dataset (Hu et al., 2025) and generate 128 responses with temperature set to 1. We calculate each response's unnormalized gradient $g_i$, along with the mean gradient $\mathbb{E}[g_i] = \frac{1}{128} \sum_{i=1}^{128} g_i$, which serves as the overall expected gradient. We then compute the squared deviation for each gradient sample $||g_i - \mathbb{E}[g_i]||^2$, the expectation of which corresponds to the gradient variance. Figure 3 illustrates this deviation for the Q, K, V projection matrices in the last layer, clearly confirming that gradient variance increases proportionally with response length. More empirical examples can be found in Appendix A.3.

During RLVR training, the response length typically grows significantly over time. Consequently, using the standard estimator $\frac{1}{G} \sum_{i=1}^{G} g_i$ without normalization would lead to increasingly high gradient variance. To mitigate this, GRPO introduces sample-level normalization, while DAPO employs batch-level length normalization, both aiming to stabilize training.

## 2.3 BIAS AND VARIANCE PROPERTIES OF EXISTING METHODS

In this subsection, we provide a detailed analysis on how different loss aggregation methods influence the bias and variance properties of the combined gradient $g$.

**Bias Properties.** We analyze the bias of each method by calculating the expectation of these estimators and comparing them with the standard policy gradient $\nabla_\theta J(\theta)$. Because $\mathbb{E}[g_i] \approx \nabla_\theta J(\theta)$, without any further assumption, we have the $\mathbb{E}[g]$ shown in Table 1.

GRPO and DAPO introduce length-dependent coefficients determined by the response lengths $L_i$, which results in a *bias* issue. As $L_i$ grows during RLVR training, these methods yield larger updates

---

[1]The variance here is strictly defined as $\text{Var}(g_i) = \mathbb{E}\left[||g_i - \mathbb{E}[g_i]||^2\right]$, which measures the expected deviation in norm and corresponds to the total variance in statistics. For simplicity, we refer to it as *variance* throughout this paper without further specification.

in the early stage and progressively smaller gradients later. For instance, the expected gradient norm per step of GRPO is $\|\mathbb{E}[\boldsymbol{g}_{\text{GRPO}}]\| = \left(\frac{1}{G}\sum_{i=1}^{G}\frac{1}{L_i}\right)\|\nabla_\theta J(\theta)\|$, where $\frac{1}{G}\sum_{i=1}^{G}\frac{1}{L_i}$ decreases as response lengths increase. Consequently, the gradient norm decreases over time, slowing convergence in the later training phase. In contrast, Dr. GRPO does not suffer from this issue, since $\mathbb{E}[\boldsymbol{g}_{\text{Dr. GRPO}}]$ always remains a constant scaling of $\nabla_\theta J(\theta)$, independent of response length.

| Method | $E(\boldsymbol{g})$ | $\text{Var}(\boldsymbol{g})$ | $\text{CV}(\boldsymbol{g})$ |
|---|---|---|---|
| GRPO | $\left(\frac{1}{G}\sum_{i=1}^{G}\frac{1}{L_i}\right)\nabla_\theta J(\theta)$ ✗ | $\frac{V}{G^2}\sum_{i=1}^{G}\frac{1}{L_i}$ | $\left(\sqrt{\sum_{i=1}^{G}\frac{1}{L_i}}\right)^{-1}\cdot\frac{\sqrt{V}}{\|\nabla_\theta J(\theta)\|}$ ↓ |
| DAPO | $\left(\frac{G}{\sum_{i=1}^{G}L_i}\right)\nabla_\theta J(\theta)$ ✗ | $\frac{V}{\sum_{i=1}^{G}L_i}$ | $\frac{\sqrt{\sum_{i=1}^{G}L_i}}{G}\cdot\frac{\sqrt{V}}{\|\nabla_\theta J(\theta)\|}$ ↑ |
| Dr. GRPO | $\frac{1}{M}\nabla_\theta J(\theta)$ ✓ | $\frac{V\sum_{i=1}^{G}L_i}{G^2 M^2}$ | $\frac{\sqrt{\sum_{i=1}^{G}L_i}}{G}\cdot\frac{\sqrt{V}}{\|\nabla_\theta J(\theta)\|}$ ↑ |
| Ours ($\alpha=1$) | $\frac{1}{M}\nabla_\theta J(\theta)$ ✓ | $\frac{V}{M^2\sum_{i=1}^{G}\frac{1}{L_i}}$ | $\left(\sqrt{\sum_{i=1}^{G}\frac{1}{L_i}}\right)^{-1}\cdot\frac{\sqrt{V}}{\|\nabla_\theta J(\theta)\|}$ ↓ |

Table 1: Comparison of $E(\boldsymbol{g})$, $\text{Var}(\boldsymbol{g})$, and $\text{CV}(\boldsymbol{g})$. ✗: biased, ✓: unbiased, ↑: high CV, ↓: low CV.

**Variance Properties.** Beyond the bias issue, a more fundamental challenge is gradient variance. On-policy methods are well known for high gradient variance due to the fact that gradients are estimated from a limited number of sampled trajectories at each update step, leading to significant estimation noise (Williams, 1992; Sutton et al., 1998; Schulman et al., 2015). In the context of RLVR, variability in response length amplify this problem, as longer responses tend to produce higher variance.

To analyze how different normalization strategies affect gradient variance, we introduce two assumptions here. First, according to the analysis in Section 2.2, we assume that the variance of each sample-level gradient $\boldsymbol{g}_i$ is proportional to its length $L_i$. Formally we assume $\text{Var}(\boldsymbol{g}_i)\approx V\cdot L_i$, where $V$ is the variance in token-level.

Second, we assume that the gradients $\boldsymbol{g}_i$ are independent across samples, as each is computed from an independently sampled trajectory $o_i$. We acknowledge that the use of group-based baselines, such as $A_i = r_i - \text{mean}(r_1,\ldots,r_G)$ or $A_i = \frac{r_i-\text{mean}(r_1,\ldots,r_G)}{\text{std}(r_1,\ldots,r_G)}$, introduces dependence between these gradients. However, this effect diminishes when the group-based baseline closely approximates the true return expectation $\mathbb{E}[r_i]$, which is only dependent on the current policy $\pi_\theta$ and question $q$.

Under the assumption that $\text{Var}(\boldsymbol{g}_i)\approx VL_i$ and all $\boldsymbol{g}_i$ are independent, the variance results are summarized in Table 1. Since $\boldsymbol{g}_{\text{GRPO}}$, $\boldsymbol{g}_{\text{DAPO}}$, and $\boldsymbol{g}_{\text{Dr. GRPO}}$ have different expectations, variance alone is not comparable, as a larger $\|\mathbb{E}(\boldsymbol{g})\|$ naturally leads to larger variance. To normalize this effect, we use the Coefficient of Variation (CV), defined as $\text{CV}(\boldsymbol{g}) = \frac{\sqrt{\text{Var}(\boldsymbol{g})}}{\|\mathbb{E}(\boldsymbol{g})\|}$, which quantifies the deviation per unit of gradient norm. We can prove $\text{CV}(\boldsymbol{g}_{\text{GRPO}}) \leq \text{CV}(\boldsymbol{g}_{\text{DAPO}}) = \text{CV}(\boldsymbol{g}_{\text{Dr. GRPO}})$ (details in Appendix A.1), implying that DAPO and Dr. GRPO induce higher relative variability than GRPO, and thus each unit of gradient norm update carries larger statistical deviation, leading to less stable optimization. Our theoretical analysis agrees with recent reports (Hong et al., 2025), showing that GRPO-style per-sample normalization is more stable than DAPO's per-token normalization.

## 3 RETHINK LOSS NORMALIZATION IN RLVR

Existing aggregation methods introduce either bias or excessive variance. We therefore ask: *Can a loss aggregation method be both unbiased and minimum-variance?* The answer is yes. We observe that this problem can be naturally reformulated within the framework of *minimum variance unbiased estimation* in statistics. Specifically, we treat the gradients obtained from responses of different lengths as independent observations of the same underlying variable (the ground-truth policy gradient), each with its own variance. Our objective is then to construct a new unbiased estimator by

optimally combining these observations so that the resulting variance is minimized. Formally, we define the problem as follows.

**Problem Definition.** Given a set of independent sample-level gradient estimators $\{g_i\}_{i=1}^G$ satisfying $\mathbb{E}[g_i] = \nabla_\theta J(\theta)$ and $\mathrm{Var}(g_i) = V L_i$, where $L_i > 0$ denotes the length associated with sample $i$ and $V$ is a constant scalar, the objective is to determine coefficients $\{x_i\}_{i=1}^G$ in the linear combination $\hat{g} = \sum_{i=1}^G x_i g_i$, such that $\mathbb{E}[\hat{g}] = \nabla_\theta J(\theta)/M$ for a given $M > 0$, while minimizing the variance $\mathrm{Var}[\hat{g}]$.

Noting that the unbiasedness constraint $\mathbb{E}[\hat{g}] = \nabla_\theta J(\theta)/M$ is equivalent to $\sum_{i=1}^G x_i = \frac{1}{M}$ and, by independence, the variance satisfies $\mathrm{Var}[\hat{g}] = \sum_{i=1}^G x_i^2 \mathrm{Var}(g_i) = V \sum_{i=1}^G L_i x_i^2$, the problem reduces to a convex optimization program which can be solved with the Lagrange multiplier method, yielding the unique minimizer: $x_i^\star = \frac{1}{M} \frac{L_i^{-1}}{\sum_{j=1}^G L_j^{-1}}$, $i = 1, \ldots, G$ (See Appendix A.4 for details).

In practice, we find it is beneficial to introduce a hyperparameter $0 \le \alpha \le 1$ to the normalization factor, which gives the following normalization weights:

$$x_i = \frac{1}{M} \frac{L_i^{-\alpha}}{\sum_{j=1}^G L_j^{-\alpha}}, \quad i = 1, \ldots, G,$$

The parameter $\alpha$ provides a tradeoff between variance reduction and utilization of long responses. While longer responses tend to introduce higher variance, sometimes they also carry richer learning signals. Choosing $\alpha < 1$ allows these signals to contribute more effectively, at the cost of increased gradient variance. We name this method *VL Norm*, as it is specially designed to match the dynamic length nature in RLVR. It has four key properties:

- **Unbiasedness:** For any $\alpha$, *VL Norm* is unbiased since $\sum_{i=1}^G x_i = \frac{1}{M}$, ensuring $\mathbb{E}[\hat{g}] = \nabla_\theta J(\theta)/M$. This preserves theoretical consistency with vanilla reinforcement learning.
- **Minimum Variance:** Choosing $\alpha = 1$ achieves the minimum possible variance under the unbiasedness constraint. Under the assumptions in Section 2.2, this is the unique solution when loss aggregation is a linear, unbiased combination.
- **Controlled Coefficient of Variation (CV):** We can show (detailed proof in Appendix A.1):
  $$\mathrm{CV}(g_{\mathrm{GRPO}}) = \mathrm{CV}(\hat{g}_{\alpha=1}) \le \mathrm{CV}(\hat{g}_{0<\alpha<1}) \le \mathrm{CV}(g_{\mathrm{DAPO}}) = \mathrm{CV}(g_{\mathrm{Dr.\,GRPO}}).$$
  Thus, *VL Norm* guarantees lower CV than DAPO and Dr. GRPO, while matching GRPO at $\alpha = 1$. When $\alpha < 1$, variance increases slightly, but long responses contribute more effectively.
- **Transition to Dr. GRPO:** Setting $\alpha = 0$ recovers the aggregation method introduced in Dr. GRPO, making it a special case of *VL Norm*.

These properties make *VL Norm* highly valuable for RLVR training. The unbiasedness property ensures consistency with standard reinforcement learning theory, preventing unexpected slowdowns caused by biased gradient estimates. Variance reduction further stabilizes training and accelerates convergence. In practice, we find that, setting $\alpha = 1$, which minimizes the variance, is a universal good choice. $\alpha = 0.75$ further increase the performance in Math, which might be due to the fact that the long response in Math task should be better utilized.

## 4 EVALUATION

### 4.1 SETTINGS

**Basics.** We evaluate the proposed *VL Norm* on two models, Qwen2.5-3B and Qwen2.5-7B (Yang et al., 2024), across two tasks: CountDown and Math. For CountDown, we train on the TinyZero dataset (Pan et al., 2025) and test on a held-out set of 1,000 samples. For Math, we train on Open Reasoner Zero dataset (Hu et al., 2025) and evaluate on MATH500 (Hendrycks et al., 2021), Minerva (Lewkowycz et al., 2022), AMC (Li et al., 2024), and AIME2024 (Li et al., 2024). The maximum response length is set to 3072, with an additional evaluation of the 3B model at length 8192. Detailed training settings and implementation details are provided in Appedix A.2 and A.5.

|  | 3B Model | | 3B Model; 8192 | | 7B Model | |
| --- | --- | --- | --- | --- | --- | --- |
| Method | Avg@8 | Pass@8 | Avg@8 | Pass@8 | Avg@8 | Pass@8 |
| GRPO Norm | 0.811 | 0.928 | 0.874 | 0.954 | **0.827** | 0.934 |
| DAPO Norm | 0.800 | 0.922 | 0.841 | 0.913 | 0.804 | 0.924 |
| Dr. GRPO Norm | 0.781 | 0.889 | 0.819 | 0.916 | 0.790 | 0.897 |
| Dr. GRPO | 0.755 | 0.879 | 0.816 | 0.923 | 0.809 | 0.924 |
| Ours | **0.847** | **0.938** | **0.898** | **0.967** | 0.821 | **0.937** |

Table 2: Detailed results on the CountDown task evaluated using Avg@8 and Pass@8.

**Baselines.** Our primary goal is to compare different loss aggregation methods under identical settings. Baselines include the aggregation methods from GRPO, DAPO, and Dr. GRPO, which we denote as *GRPO Norm*, *DAPO Norm*, and *Dr. GRPO Norm*, respectively. For *VL Norm*, we mainly set $\alpha = 1$ for CountDown and $\alpha = 0.75$ for Math. The impact of different $\alpha$ values is further analyzed in Section 4.4. To ensure fairness, we adopt the same advantage estimator, $A_i = \frac{r_i - \text{mean}(\{r_1, r_2, ..., r_G\})}{\text{std}(\{r_1, r_2, ..., r_G\})}$, for GRPO Norm, DAPO Norm, Dr. GRPO Norm, and *VL Norm*. This differs from the original Dr. GRPO, which uses $A_i = r_i - \text{mean}(\{r_1, r_2, \ldots, r_G\})$ (Liu et al., 2025). Therefore, we also include the original Dr. GRPO as an additional baseline. Furthermore, we acknowledge that DAPO incorporates additional techniques beyond loss aggregation to mitigate the impact of lengthy responses, such as overlong response filtering and soft punishment. It also introduces dynamic sampling to improve training performance. To compare our method and these techniques, we provide further experiments in Section 4.3.

## 4.2 MAIN RESULTS

**Training Dynamics.** Figure 2 presents the training dynamics across all 6 settings, while Figure 5 highlights 2 representative settings for better readability. For the CountDown task, the score is measured by Avg@8 with temperature = 1 on the held-out test subset. For the Math task, we report a weighted average (weighted by the question number) of Avg@8 scores across four test datasets.

Across nearly all settings, the proposed *VL Norm* outperforms all baselines, achieving both higher stability during training and superior test scores. The only exception is the CountDown 7B setting, where GRPO Norm slightly surpasses our method, though *VL Norm* still shows clear advantages over the other three baselines. We find the proposed method exhibits highly monotonicity of performance improvement. To quantify this, we introduce a "monotonicity score", defined as the Pearson correlation between $\{0, 1, 2, \ldots\}$ and the test scores $\{score_0, score_{50}, score_{100}, \ldots\}$ recorded every 50 steps. As shown in Table 5, our method achieves the highest average correlation, with values consistently above $0.94$ across all settings. This demonstrates that *VL Norm* promotes a stable and steadily improving training dynamics.

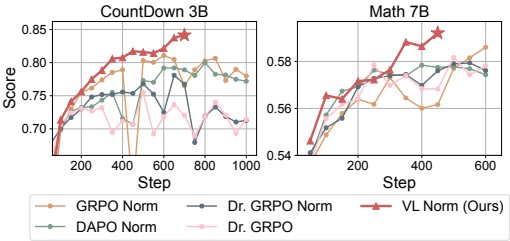
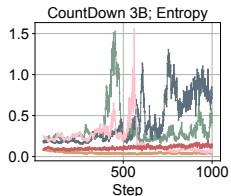
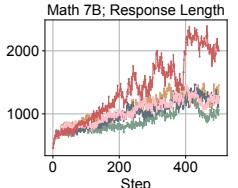

Figure 5: Selected training dynamics. Please refer to Figure 2 for training dynamics on all settings.

Figure 6: Entropy on CountDown and 3B model. Legend is the same as Figure 5.

Figure 7: Response length on Math and 7B model. Legend is the same as Figure 5.

**CountDown.** We report the best Avg@8 and Pass@8 metrics throughout training of each method in Table 2 for the CountDown task. Overall, the proposed method achieves the strongest performance, with GRPO Norm being the second best. We find that LLMs often generate responses exceeding the maximum response length in the CountDown task, which tends to dominate the gradient and increase its variance. The low-CV property shared by GRPO Norm and our method helps

alleviate this problem and stabilize training. However, GRPO introduces a bias issue in gradient estimation, which may hinder convergence in the later stage of training. As shown in Figure 5, both our method and GRPO Norm converge quickly within the first 300 steps, but GRPO Norm becomes less effective as training progresses, while our method continues to converge at a fast rate. We observe explicit entropy difference caused by different aggregation method. During training, Dr. GRPO Norm, Dr. GRPO, and DAPO Norm typically exhibit entropy spikes, which are often associated with sudden drops in performance, as shown in Figure 5 and Figure 6, potentially due to their high-CV property. In contrast, the proposed method consistently maintains an entropy between 0.1 and 0.2, which is beneficial for stable training.

**Math.** For Math task, we present the performance of the models achieving the best weighted average Avg@8 score across the four test datasets in Table 4. Our method consistently outperforms all baselines, achieving the highest weighted average score as well as the highest average score. We observe that the proposed method can lead to sudden increases in response length during training, which are closely associated with performance improvements. As shown in Figure 5 and Figure 7, for the 7B model, the response length exhibits sharp increases around 300 and 400 steps, coinciding with noticeable boosts in test accuracy. However, unlike in the CountDown task, we do not observe significant differences in entropy.

## 4.3 COMBINATION WITH FULL DAPO

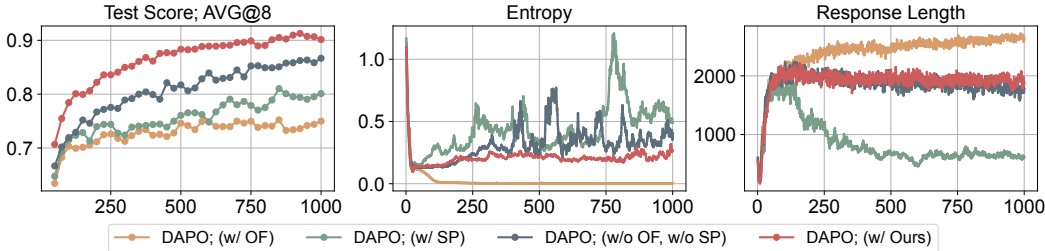

Figure 8: Comparison between our methods and full DAPO on CountDown task and 3B model.

We acknowledge DAPO introduces two additional techniques to mitigate the high variance issues of long responses: Overlong filtering and Soft punishment (Yu et al., 2025). To understand their effects, we experiment with the full DAPO method, including Dynamic sampling, Clip higher, Per-token loss, and either Overlong filtering or Soft punishment. We conduct the following experiments:

- **DAPO w/ OF**: Full DAPO with Overlong filtering.
- **DAPO w/ SP**: Full DAPO with Soft punishment. The punishment interval is set to 2048.
- **DAPO w/o OF, w/o SP**: Dynamic sampling, clip higher, and per-token loss only.
- **DAPO w/ Ours**: Dynamic sampling and clip higher, with *VL Norm* ($\alpha = 1$).

Figure 8 shows the training dynamics. **DAPO w/ Ours** clearly outperforms all baselines, achieving the best Avg@8 of 0.913. In Table 2, *VL Norm* achieves the 0.847. The improvement from 0.847 to 0.913 comes from Dynamic sampling, which is the only difference. Neither overlong filtering nor soft punishment is effective in this setting. Overlong filtering produces the longest responses, likely because overlong responses receive no penalty once masked. Soft punishment yields the shortest responses, which limits both exploration and performance. These observations align with recent works (Du et al., 2025; Shang et al., 2025). In comparison, *VL Norm* provides a simpler and more unified approach to handle long responses. It also stabilizes entropy, consistent with the findings in subsection 4.2.

## 4.4 CHOICE OF HYPERPARAMETERS

In this section, we investigate the hyperparameter $0 \leq \alpha \leq 1$ in *VL Norm*. Since each gradient is scaled by $L_i^{-\alpha}$, larger $\alpha$ weakens the contribution of long responses in the aggregated gradient

| Method | CD; 3B | CD; 7B | Math; 3B | Math; 7B | AVG |
|---|---|---|---|---|---|
| GRPO Norm | 0.811 | **0.827** | 0.509 | 0.573 | 0.680 |
| DAPO Norm | 0.800 | 0.804 | 0.505 | 0.578 | 0.672 |
| Dr. GRPO Norm | 0.781 | 0.790 | 0.502 | 0.576 | 0.662 |
| Dr. GRPO | 0.755 | 0.809 | 0.506 | 0.579 | 0.662 |
| Ours; $\alpha$=0.5 | 0.805 | 0.825 | 0.505 | 0.586 | 0.680 |
| Ours; $\alpha$=0.75 | 0.838 | 0.797 | **0.517** | **0.592** | 0.686 |
| Ours; $\alpha$=1.0 | **0.847** | 0.821 | 0.506 | 0.590 | **0.691** |

Table 3: Performance comparison across different hyperparameters ($\alpha = 0.5, 0.75, 1.0$). Cells highlighted in light green denote results surpassing at least three out of the four baselines, while those in dark green denote results surpassing all baselines.

but results in smaller gradient variance. $\alpha = 1$ achieves the minimum variance, while $0 < \alpha < 1$ increases variance but leverages long responses more effectively. Setting $\alpha = 0$ is equivalent to the aggregation method proposed in Dr. GRPO. We evaluate $\alpha = 0.5, 0.75, 1.0$ on four settings (Math, CountDown; 3B and 7B), with results shown in Table 3. For clarity, we mark parameter choices in light green if they outperform at least three baselines and in dark green if they surpass all baselines. Importantly, all three tested values ($\alpha = 0.5, 0.75, 1.0$) outperform the baselines on four tasks' average. On Math, $\alpha = 0.75$ performs better than $\alpha = 1$, likely because longer responses are more informative and should be better utilized. Overall, $\alpha = 1$ usually gives good results.

## 5 RELATED WORKS

**Loss Aggregation Methods in RLVR**   RLVR methods differ in how they aggregate the policy loss. GRPO applies a length-dependent normalizer to each sample (Shao et al., 2024); Dr. GRPO removes this term and normalize the gradient with a constant (Liu et al., 2025); DAPO uses a batch-level length-dependent normalizer (Yu et al., 2025). We show that the length-dependent normalizers in GRPO and DAPO have bias in gradient estimating, while the constant normalizer in Dr. GRPO incurs high variance. To address both issues, we propose *VL Norm*, an aggregation method that is unbiased and minimizes variance.

**Classical Gradient Variance Reduction Methods.**   Gradient variance is a central challenge in classical policy-gradient methods. REINFORCE reduces variance by subtracting a baseline from returns (Williams, 1992). Actor–critic methods extend this idea by learning a value-function baseline (Sutton et al., 1998). GAE further stabilizes learning by interpolating between Monte Carlo and low-variance temporal-difference returns (Schulman et al., 2015). While these techniques primarily address variance from rewards, RLVR introduces an additional variance source: long and variable trajectory lengths in reasoning tasks. Our method provides a loss-aggregation scheme that aligns with classical RL and explicitly minimizes gradient variance, yielding strong empirical results.

**Length-dependent Rewards in RLVR.**   Several works make the reward depend on response length to mitigate the overthinking problem. Kimi K1.5 adds additional rewards for short correct answers and penalties for long incorrect ones (Team et al., 2025). GRPO-LEAD avoids penalizing incorrect answers and applies an exponential, length-based reward only to correct ones (Zhang & Zuo, 2025). ShortRL introduces a neutral-length zone that neither rewards nor penalizes moderately long responses to preserve diversity (Yuan et al., 2025). Our method leaves the reward unchanged and is therefore orthogonal to these designs.

## 6 CONCLUSION

In this paper, we introduce *VL Norm*, a loss-aggregation method tailored to the high length variability in RLVR. It yields an unbiased policy-gradient estimator consistent with vanilla RL theory and minimizes gradient variance, enabling more stable training and convergence to stronger models. Empirically, *VL Norm* consistently outperforms alternatives across model sizes, maximum response lengths, and tasks.

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

## A APPENDIX

### A.1 PROOF OF MAGNITUDE RELATION OF CV VALUES

We have the following CV values for each method:

$$\text{CV}(\boldsymbol{g}_{\text{GRPO}}) = \frac{1}{\sqrt{\sum_{i=1}^{G} \frac{1}{L_i}}} \cdot \frac{\sqrt{V}}{\|\nabla_\theta J(\theta)\|},$$

$$\text{CV}(\boldsymbol{g}_{\text{DAPO}}) = \frac{\sqrt{\sum_{i=1}^{G} L_i}}{G} \cdot \frac{\sqrt{V}}{\|\nabla_\theta J(\theta)\|},$$

$$\text{CV}(\boldsymbol{g}_{\text{Dr.GRPO}}) = \frac{\sqrt{\sum_{i=1}^{G} L_i}}{G} \cdot \frac{\sqrt{V}}{\|\nabla_\theta J(\theta)\|},$$

$$\text{CV}(\boldsymbol{g}_{\text{Ours}}) = \frac{\sqrt{\sum_{i=1}^{G} L_i^{1-2\alpha}}}{\sum_{j=1}^{G} L_j^{-\alpha}} \cdot \frac{\sqrt{V}}{\|\nabla_\theta J(\theta)\|}$$

In this subsection, we provide proof of their magnitude relation.

**Proof of** $\text{CV}(\boldsymbol{g}_{\text{GRPO}}) \leq \text{CV}(\boldsymbol{g}_{\text{DAPO}})$. It is equal to prove:

$$\frac{1}{\sqrt{\sum_{i=1}^{G} \frac{1}{L_i}}} \leq \frac{\sqrt{\sum_{i=1}^{G} L_i}}{G}, \quad \text{with } L_i > 0.$$

This is equivalent to

$$G \leq \sqrt{\left(\sum_{i=1}^{G} L_i\right)\left(\sum_{i=1}^{G} \frac{1}{L_i}\right)} \iff G^2 \leq \left(\sum_{i=1}^{G} L_i\right)\left(\sum_{i=1}^{G} \frac{1}{L_i}\right).$$

By the Cauchy–Schwarz inequality applied to $\left(\sqrt{L_1}, \ldots, \sqrt{L_G}\right)$ and $\left(1/\sqrt{L_1}, \ldots, 1/\sqrt{L_G}\right)$, we have:

$$\left(\sum_{i=1}^{G} \sqrt{L_i} \cdot \frac{1}{\sqrt{L_i}}\right)^2 \leq \left(\sum_{i=1}^{G} L_i\right)\left(\sum_{i=1}^{G} \frac{1}{L_i}\right),$$

whose left-hand side equals $G^2$. Hence the desired inequality holds. Equality occurs if and only if all $L_i$ are equal.

**Proof of** $\mathrm{CV}(\boldsymbol{g}_{\mathrm{GRPO}}) \leq \mathrm{CV}(\boldsymbol{g}_{\mathrm{Ours}})$**.**  It is equal to prove:

$$\frac{1}{\sqrt{\sum_{i=1}^{G} \frac{1}{L_i}}} \leq \frac{\sqrt{\sum_{i=1}^{G} L_i^{1-2\alpha}}}{\sum_{j=1}^{G} L_j^{-\alpha}}, \qquad L_i > 0.$$

Since all terms are positive, this is equivalent to

$$\left(\sum_{i=1}^{G} L_i^{-\alpha}\right)^2 \leq \left(\sum_{i=1}^{G} L_i^{1-2\alpha}\right)\left(\sum_{i=1}^{G} \frac{1}{L_i}\right).$$

Apply the Cauchy–Schwarz inequality to $\left(L_1^{\frac{1-2\alpha}{2}}, \ldots, L_G^{\frac{1-2\alpha}{2}}\right)$ and $\left(L_1^{-\frac{1}{2}}, \ldots, L_G^{-\frac{1}{2}}\right)$:

$$\left(\sum_{i=1}^{G} L_i^{\frac{1-2\alpha}{2}} \cdot L_i^{-\frac{1}{2}}\right)^2 \leq \left(\sum_{i=1}^{G} L_i^{1-2\alpha}\right)\left(\sum_{i=1}^{G} \frac{1}{L_i}\right).$$

The left-hand side simplifies to $\left(\sum_{i=1}^{G} L_i^{-\alpha}\right)^2$, which yields the desired inequality. Equality holds for all $\{L_i\}$ when $\alpha = 1$, and for $\alpha \neq 1$ only when all $L_i$ are equal.

**Proof of** $\mathrm{CV}(\boldsymbol{g}_{\mathrm{Ours}}) \leq \mathrm{CV}(\boldsymbol{g}_{\mathrm{DAPO}})$ **for** $\alpha \in [0,1]$**.**  If $\alpha = 0$, then $\mathrm{CV}(\boldsymbol{g}_{\mathrm{Ours}}) = \mathrm{CV}(\boldsymbol{g}_{\mathrm{DAPO}})$. If $\alpha = 1$, then $\mathrm{CV}(\boldsymbol{g}_{\mathrm{Ours}}) = \mathrm{CV}(\boldsymbol{g}_{\mathrm{GRPO}}) \leq \mathrm{CV}(\boldsymbol{g}_{\mathrm{DAPO}})$.

For $\alpha \in [0,1]$, define

$$\Phi(t) := \ln\left(\sum_{i=1}^{G} L_i^t\right), \qquad f(\alpha) := \ln\left(\frac{\sqrt{\sum_{i=1}^{G} L_i^{1-2\alpha}}}{\sum_{i=1}^{G} L_i^{-\alpha}}\right) = \tfrac{1}{2}\,\Phi(1-2\alpha) - \Phi(-\alpha).$$

It is well known that the *log-sum-exp* function is convex; hence $\Phi$ is convex on $\mathbb{R}$, and therefore $\Phi'$ is nondecreasing on $\mathbb{R}$.

Differentiating $f$ gives

$$f'(\alpha) = -\Phi'(1-2\alpha) + \Phi'(-\alpha).$$

Since $-\alpha \leq 1 - 2\alpha$ for all $\alpha \in [0,1]$ and $\Phi'$ is nondecreasing, we obtain

$$\Phi'(-\alpha) \leq \Phi'(1-2\alpha) \quad \Longrightarrow \quad f'(\alpha) \leq 0.$$

Thus $f$ is nonincreasing on $[0,1]$, so $f(\alpha) \leq f(0)$:

$$\frac{\sqrt{\sum_{i=1}^{G} L_i^{1-2\alpha}}}{\sum_{i=1}^{G} L_i^{-\alpha}} \leq \frac{\sqrt{\sum_{i=1}^{G} L_i}}{G},$$

which is exactly $\mathrm{CV}(\boldsymbol{g}_{\mathrm{Ours}}) \leq \mathrm{CV}(\boldsymbol{g}_{\mathrm{DAPO}})$ for all $\alpha \in [0,1]$. Equality holds at $\alpha = 0$; for $\alpha \in (0,1]$, equality occurs if and only if all $L_i$ are equal.

Overall, we have the following magnitude order:

$$\mathrm{CV}(\boldsymbol{g}_{\mathrm{GRPO}}) = \mathrm{CV}(\boldsymbol{g}_{\mathrm{Ours};\,\alpha=1}) \leq \mathrm{CV}(\boldsymbol{g}_{\mathrm{Ours};\,0<\alpha<1}) \leq \mathrm{CV}(\boldsymbol{g}_{\mathrm{DAPO}}) = \mathrm{CV}(\boldsymbol{g}_{\mathrm{Dr.\,GRPO}}).$$

| Method | Minerva | AIME2024 | MATH500 | AMC | W.Average | Average |
|---|---|---|---|---|---|---|
| *3B Model* | | | | | | |
| GRPO Norm | 0.228 | 0.088 | 0.709 | 0.377 | 0.509 | 0.350 |
| DAPO Norm | **0.231** | 0.058 | 0.703 | 0.369 | 0.505 | 0.340 |
| Dr. GRPO Norm | 0.224 | **0.121** | 0.697 | 0.383 | 0.502 | 0.356 |
| Dr. GRPO | **0.231** | 0.088 | 0.699 | 0.392 | 0.506 | 0.352 |
| Ours | 0.228 | 0.104 | **0.720** | **0.395** | **0.517** | **0.362** |
| *3B Model; 8192 Max Response Length* | | | | | | |
| GRPO Norm | 0.233 | 0.096 | 0.703 | 0.378 | 0.508 | 0.353 |
| DAPO Norm | 0.227 | 0.104 | 0.699 | **0.399** | 0.506 | 0.357 |
| Dr. GRPO Norm | 0.221 | 0.096 | 0.691 | 0.386 | 0.498 | 0.348 |
| Dr. GRPO | 0.226 | 0.100 | 0.699 | 0.383 | 0.503 | 0.352 |
| Ours | **0.239** | **0.121** | **0.704** | 0.370 | **0.510** | **0.359** |
| *7B Model* | | | | | | |
| GRPO Norm | 0.275 | 0.117 | 0.777 | **0.488** | 0.573 | 0.414 |
| DAPO Norm | 0.290 | 0.158 | 0.778 | 0.474 | 0.578 | 0.425 |
| Dr. GRPO Norm | 0.299 | 0.163 | 0.769 | 0.471 | 0.576 | 0.426 |
| Dr. GRPO | 0.295 | 0.142 | 0.777 | 0.476 | 0.579 | 0.422 |
| Ours | **0.302** | **0.167** | **0.794** | 0.483 | **0.592** | **0.436** |

Table 4: Detailed results on the Math task. Performance is evaluated using Avg@8 on each dataset. W.Average denotes the weighted average of the four Avg@8 scores by question counts, and Average is the direct mean. Across three settings, our method consistently yields the best W.Average and Average.

## A.2 TRAINING DETAILS

We provide additional details of the reinforcement learning setup. All experiments are conducted with a batch size of 1280, consisting of 128 prompts with 10 rollouts per prompt. The mini-batch size is 320 and the learning rate is $1 \times 10^{-6}$. In both tasks, we use a binary reward: 1 for success and 0 for failure. The maximum response length is set to 3072 for all models and tasks, and we further evaluate the 3B model with response length 8192 to test robustness. Following previous works, we do not use KL loss or entropy loss (Yu et al., 2025). We adopt the clip higher trick, setting the lower clip ratio to 0.2 and the upper clip ratio to 0.3 (Yu et al., 2025). For Dr. GRPO and our method, $M$ is set to the maximum response length, consistent with Liu et al. (2025).

For Math, we train the 7B model for 1 epoch, the 3B model for 3 epochs, the 3B model with length 8192 for 2 epochs, and we evaluate the model on MATH500, Minerva, AMC, and AIME2024 every 50 steps. For CountDown, we train for 500 steps with the 7B model, and for 1000 steps with the 3B models under both length settings (3072 and 8192), and the model is evaluated on a held-out test set of 1000 samples every 50 steps.

For both Math and Countdown, we use Avg@8 accuracy with the sampling temperature fixed at 1 as the evaluation metric. The training dynamics shown in Figures 2, 5, and 8 are based on Avg@8 accuracy evaluated every 50 steps. For Countdown, this corresponds directly to Avg@8 accuracy on its test set, while for Math, it is computed as a weighted average over four test datasets, where the weights are proportional to the number of questions in each dataset.

The results in Tables 2 and 4 are reported for the best model under each setting. The best model is selected by the highest Avg@8 for Countdown or the highest weighted Avg@8 for Math, and this criterion is applied consistently across all methods. Table 5 further characterizes the training process using a monotonicity score, defined as the Pearson correlation between the step indices $\{0, 1, 2, \dots\}$ and the corresponding Avg@8 scores $\{score_0, score_{50}, score_{100}, \dots\}$ evaluated every 50 steps.

## A.3 ADDITIONAL EXAMPLES ON GRADIENT VARIANCE AND RESPONSE LENGTH

In Section 2.2, we presented an empirical example on the relation between gradient variance and response length. Specifically, we sampled one math question from the Open Reasoner Zero dataset (Hu et al., 2025) and generated 128 responses with temperature 1. For each response, we computed

| Method | CD; 3B | CD; 3B; 8192 | CD; 7B | Math; 3B | Math; 3B; 8192 | Math; 7B |
|--------|--------|--------------|--------|----------|----------------|----------|
| GRPO Norm | 0.652 | 0.726 | 0.743 | 0.759 | 0.975 | 0.818 |
| DAPO Norm | 0.804 | 0.902 | 0.882 | 0.854 | 0.900 | 0.748 |
| Dr. GRPO Norm | 0.134 | 0.736 | 0.736 | 0.904 | 0.867 | 0.937 |
| Dr. GRPO | 0.171 | 0.581 | **0.970** | 0.954 | 0.942 | 0.755 |
| Ours | **0.986** | **0.992** | 0.948 | **0.974** | **0.992** | **0.967** |

Table 5: Monotonicity score, defined as the Pearson correlation between $\{0, 1, 2, \dots\}$ and the test scores $\{score_0, score_{50}, score_{100}, \dots\}$ recorded every 50 steps. Our method exhibits the highest monotonicity score overall, indicating it promotes steady improvement between training steps.

its unnormalized gradient $\boldsymbol{g}_i$ and the mean gradient $\mathbb{E}[\boldsymbol{g}_i] = \frac{1}{128}\sum_{i=1}^{128}\boldsymbol{g}_i$. The squared deviation $||\boldsymbol{g}_i - \mathbb{E}[\boldsymbol{g}_i]||^2$ was then used to estimate the gradient variance.

Here, we provide additional examples. Figure 9 shows the squared gradient deviation of QKV projections in the last layer for two more samples, while Figure 10 reports the same analysis on QKV and MLP projections in the 14th layer. In all cases, the squared deviation grows approximately linearly with response length. Since the expectation of the squared deviation corresponds to variance, these results further support our claim that gradient variance scales linearly with response length.

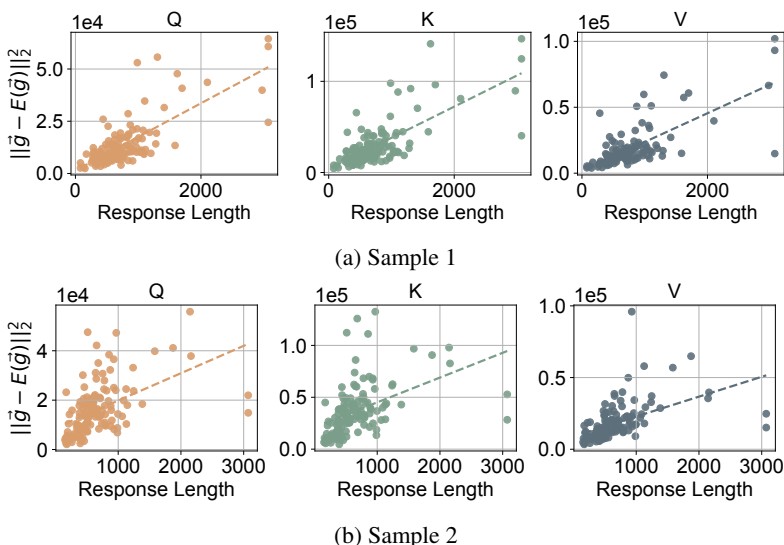

(a) Sample 1

(b) Sample 2

Figure 9: Deviation $||\boldsymbol{g}_i - \mathbb{E}[\boldsymbol{g}_i]||^2$ for two randomly selected samples on the Q, K, V projection in the last layer. $\mathbb{E}[\boldsymbol{g}_i]$ is estimated by the average of 128 rollouts for each sample.

### A.4 LAGRANGE MULTIPLIER TO FIND *VL-Norm*

Following Section 3, we formulate the loss aggregation problem. Given a set of independent sample-level gradient estimators $\{g_i\}_{i=1}^{G}$ with $\mathbb{E}[g_i] = \nabla_\theta J(\theta)$, $\text{Var}(g_i) = VL_i$, where $L_i > 0$ is the response length of sample $i$, we aim to construct a linear combination $\hat{g} = \sum_{i=1}^{G} x_i g_i$ that satisfies the unbiasedness constraint $\mathbb{E}[\hat{g}] = \frac{1}{M}\nabla_\theta J(\theta)$ while minimizing its variance $\text{Var}(\hat{g}) = \sum_{i=1}^{G} x_i^2 \text{Var}(g_i)$.

Hence, the optimization problem is

$$\min_{\{x_i\}} \quad V\sum_{i=1}^{G} L_i x_i^2 \quad \text{s.t.} \quad \sum_{i=1}^{G} x_i = \frac{1}{M}.$$

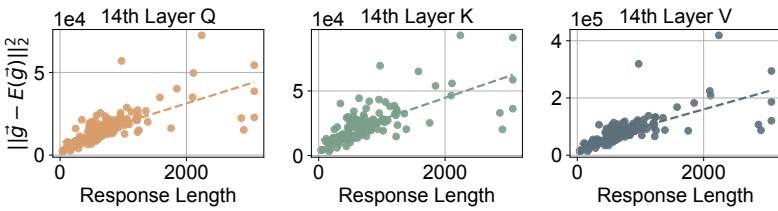

(a) Gradient deviation of Q, K, V projection in the 14th layer.

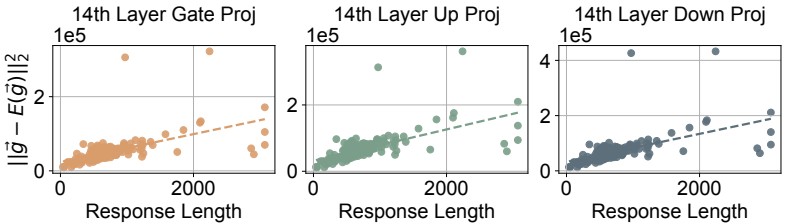

(b) Gradient deviation of MLP projection in the 14th layer.

Figure 10: GradientDeviation $||\boldsymbol{g}_i - \mathbb{E}[\boldsymbol{g}_i]||^2$ for a randomly selected samples in the QKV projection and MLP projection in the 14th layer. $\mathbb{E}[\boldsymbol{g}_i]$ is estimated by the average of 128 rollouts for each sample.

The Lagrangian is

$$\mathcal{L}(x, \lambda) = V \sum_{i=1}^{G} L_i x_i^2 + \lambda \left( \sum_{i=1}^{G} x_i - \frac{1}{M} \right).$$

Taking derivatives with respect to $x_i$ gives

$$\frac{\partial \mathcal{L}}{\partial x_i} = 2V L_i x_i + \lambda = 0 \quad \Rightarrow \quad x_i = -\frac{\lambda}{2V L_i}.$$

Applying the constraint $\sum_{i=1}^{G} x_i = \frac{1}{M}$, we obtain

$$\sum_{i=1}^{G} x_i = -\frac{\lambda}{2V} \sum_{i=1}^{G} \frac{1}{L_i} = \frac{1}{M}$$

which leads to

$$\lambda = -\frac{2V}{M} \left( \sum_{j=1}^{G} \frac{1}{L_j} \right)^{-1}.$$

Substituting back yields

$$x_i^\star = -\frac{\lambda}{2V L_i} = \frac{1}{M} \cdot \frac{L_i^{-1}}{\sum_{j=1}^{G} L_j^{-1}}, \qquad i = 1, \ldots, G.$$

This provides the unique unbiased minimum-variance solution.

### A.5   IMPLEMENTATION OF *VL-Norm*

We provide the following `PyTorch` implementation of *VL-Norm*. For clarity, we omit importance sampling and clipping.

```python
def vl_norm(advantages, lengths, log_prob, log_prob_mask, max_length, alpha):
```

```
# apply 1/M normalization; M is default to max_length
masked_log_prob = log_prob * log_prob_mask
sum_log_prob_per_sample = masked_log_prob.sum(dim=1) / float(max_length)
sample_losses = -advantages * sum_log_prob_per_sample

# use length mean for numerical stability
lengths_norm = lengths / lengths.mean()

# apply VL Norm aggregation
inv = lengths_norm ** (-alpha)
w = inv / inv.mean()
loss = (w * sample_losses).mean()
return loss
```

Here, `advantages` and `lengths` are tensors of shape $(N,)$, representing the advantages and sequence lengths of $N$ samples. `log_prob` and `log_prob_mask` are tensors of shape $(N, \text{max\_length})$, corresponding to the token-level log-probabilities and their masks.

For numerical stability, we make two modifications to the original formula. First, instead of directly computing $L_i^{-\alpha}$, we normalize the lengths by the average length $\bar{L}$ and use $(L_i/\bar{L})^{-\alpha}$. Second, the original *VL-Norm* should be `w = inv / inv.sum()` and `loss = (w * sample_losses).sum()`, which is numerically equivalent to `w = inv / inv.mean()` and `loss = (w * sample_losses).mean()`. We adopt the latter form since it is more stable in practice.

## A.6 EXPERIMENTS ON LLAMA MODELS

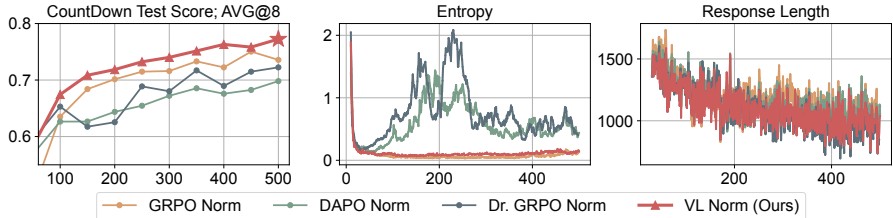

Figure 11: Training Dynamics on CountDown task with the Llama-3.2-3B-Instruct model.

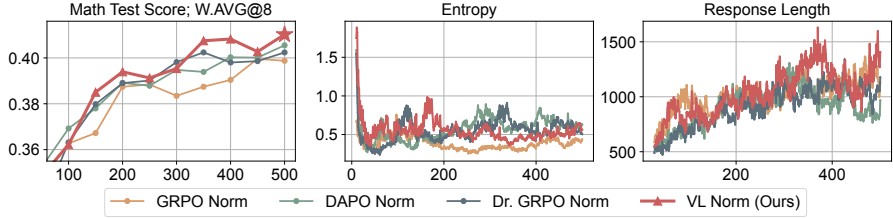

Figure 12: Training Dynamics on Math task with the Llama-3.2-3B-Instruct model.

We further evaluate our method on the Llama-3.2-3B-Instruct model (Dubey et al., 2024) using both the CountDown and Math tasks. The model is trained for 500 steps with a maximum response length of 3072, and we set $\alpha = 1$ for the proposed *VL Norm*. All other training configurations follow those used for the Qwen models. Figure 11 and Figure 12 present the full training dynamics, including testing scores (evaluated every 50 steps), entropy, and response length.

Across both tasks, *VL Norm* consistently outperforms the baseline methods (GRPO Norm, DAPO Norm, and Dr. GRPO Norm). On CountDown, our method achieves higher test scores at every evaluation point from step 100 to step 500. For Math, while the improvements are more modest, *VL Norm* remains the best-performing method from step 350 onward.

We also make several observations regarding entropy and response length. In CountDown, *VL Norm* and GRPO Norm maintain relatively stable entropy, whereas DAPO Norm and Dr. GRPO Norm fluctuate more significantly, which is consistent with our experiments on Qwen models. In Math, the entropy changes are similar across methods, but *VL Norm* produces slightly longer responses in the middle stage of training, which may be related to its stronger performance.

## A.7 FULL TRAINING DYNAMICS

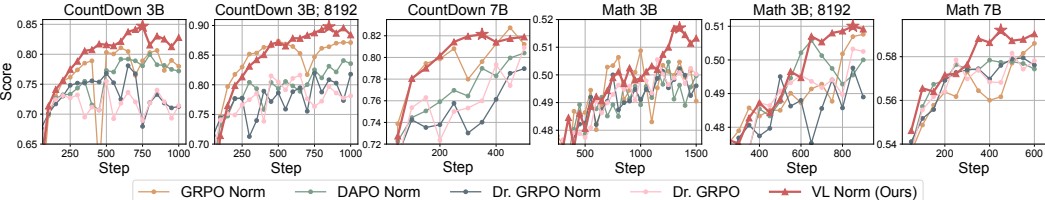

Figure 13: Full training dynamics of *VL Norm* compared with baselines across tasks (CountDown, Math), maximum lengths (3072, 8192), and model sizes (3B, 7B).

For completeness, Figure 13 presents the full training curves across all six settings. Overall, *VL Norm* consistently outperforms all baseline methods. The only exception occurs in the CountDown 7B setting, where GRPO Norm slightly surpasses our method. This does not substantially weaken our conclusion for three reasons: (1) *VL Norm* still shows clear improvements over the other three baselines; (2) the score difference between *VL Norm* and GRPO Norm is small (0.821 vs. 0.827); and (3) *VL Norm* reaches this accuracy roughly 100 steps earlier than GRPO Norm.

## A.8 LARGE-SCALE VALIDATION OF THE ASSUMPTION IN 2.2

Our method is based on a core assumption that the variance of the unnormalized gradient grows approximately linearly with the response length, as proposed in Section 2.2. To further validate this assumption, we perform a large-scale empirical study. We sample 1024 problems from Open Reasoner Zero dataset (Hu et al., 2025) and generate 128 responses for each problem, resulting in a total of 131,072 responses. For each problem, we approximate the ground-truth gradient by averaging the gradients of its 128 responses, and for each response we compute its gradient deviation. This deviation is evaluated on the Q projection, K projection, and the first MLP projection in the last layer of Qwen2.5-7B and Llama-3.2-3B-Instruct. After collecting all measurements, we compute the Pearson correlation between response length and gradient deviation across all 131,072 samples, yielding the results shown in Table 6.

| Model | Q Proj | K Proj | MLP Proj |
|---|---|---|---|
| Llama-3.2-3B-Instruct | 0.788 | 0.800 | 0.907 |
| Qwen2.5-7B | 0.740 | 0.714 | 0.683 |

Table 6: Pearson correlation between response length and gradient deviation across different parameter types.

Across different models and parameter types, the Pearson correlation ranges from 0.68 to 0.91, indicating a strong positive relationship between response length and gradient deviation. A linear model that predicts deviation from response length yields an $R^2$ between 0.5 and 0.8, demonstrating a medium to strong linear dependence. These large-scale empirical findings provide strong support for our central assumption that the variance of the unnormalized gradient grows approximately linearly with the response length.

Beyond validating the linear growth assumption, we further quantify the actual variance reduction achieved by our method. Specifically, we compute the coefficient of variation (CV) for different parameters and compare our method against DAPO Norm and Dr. GRPO Norm.

| Model | Q Proj | K Proj | MLP Proj |
|---|---|---|---|
| Llama-3.2-3B-Instruct | $-36.8\%$ | $-34.7\%$ | $-31.4\%$ |
| Qwen2.5-7B | $-22.8\%$ | $-22.0\%$ | $-16.9\%$ |

Table 7: Relative reduction in coefficient of variation (CV) achieved by our method compared with DAPO Norm and Dr. GRPO Norm across different parameters.

Table 7 reports the percentage reduction in CV achieved by our method. *VL Norm* consistently reduces the CV by $16.9\% - 36.8\%$ across Q, K, and MLP projections. This substantial decrease in CV demonstrates that *VL Norm* effectively reduces gradient variance in practice.

### A.9 THE USE OF LLMS

We acknowledge the use of large language models to refine the writing in this paper, and we carefully reviewed all model-generated content to ensure precise expression and appropriateness.

