# OpenReview forum: "VL Norm: Rethink Loss Aggregation in RLVR"
_ICLR.cc/2026/Conference — Submitted to ICLR 2026_

### Official Review · Reviewer_Z4eq · 2025-10-23

**Soundness:** 3
**Presentation:** 3
**Contribution:** 3
**Rating:** 4
**Confidence:** 3

**Summary:**

This paper proposes VL Norm, a new loss aggregation method designed to reduce the gradient variance caused by variable response lengths in reinforcement learning with verifiable rewards (RLVR). The method reformulates the gradient aggregation problem as a minimum variance unbiased estimation (MVUE) problem and introduces a tunable parameter α to control the trade-off between variance and utilization of long responses. Empirical results on tasks such as CountDown and Math using Qwen2.5 models are presented, showing that VL Norm outperforms prior approaches like GRPO, DAPO, and Dr. GRPO in stability and final accuracy.

**Strengths:**

1. The formulation of loss aggregation as an MVUE problem is mathematically elegant and conceptually interesting.
2. The method generalizes several prior approaches (e.g., GRPO, Dr. GRPO) as special cases of VL Norm via the α parameter, which adds flexibility.
3. The paper provides comprehensive experiments across multiple tasks, model sizes, and maximum response lengths.

**Weaknesses:**

1. Lack of standard deviation or variance reporting, despite the paper’s core focus on variance reduction. In Table 2, Table 3, and Table 4, performance metrics such as Avg@8 and Pass@8 are reported without any standard deviation or confidence interval:
“Table 2: Detailed results on the CountDown task…”
“Table 3: Performance comparison across different hyperparameters…”
This makes it impossible to assess whether the performance improvements are statistically significant, especially when the observed differences are small (e.g., Math task in Table 4).

2. GRPO Norm appears to achieve similar entropy stability, which undermines VL Norm’s claimed benefit in stabilizing training. The paper states: “VL Norm consistently maintains an entropy between 0.1 and 0.2, which is beneficial for stable training.” (p. 7, §4.2)
However, in Figure 6, GRPO Norm shows comparable entropy behavior, suggesting that low entropy is not unique to VL Norm.

3. VL Norm may exacerbate length bias, contrary to its motivation. In Figure 7, VL Norm exhibits a sharp increase in response length (especially in Math 7B), unlike the baselines. Yet the paper interprets this as a positive signal:
“The response length exhibits sharp increases…coinciding with noticeable boosts in test accuracy.” (p. 7)
This suggests that longer responses are being disproportionately rewarded, which could reinforce verbosity bias—a known issue in reward modeling. This contradicts the motivation to control length-induced gradient variance (p. 2–3).

**Questions:**

1. In Figure 7, VL Norm results in longer responses, which seems to contradict the paper’s motivation of mitigating length effects. How does the method avoid reinforcing verbosity bias, especially given prior work (e.g., BSR) that explicitly addresses this issue?
2. In Figure 6, GRPO Norm maintains similarly low entropy, yet the paper presents this as a unique property of VL Norm. Could the authors clarify whether VL Norm has distinct stability benefits beyond what GRPO already offers?
3. The paper states:
“Extensive experiments show that it consistently achieves superior results…” (Abstract)
Yet in CountDown 7B (Table 2), GRPO Norm actually outperforms VL Norm. Why is this not discussed?

---

> ### Author Response · Authors · 2025-11-19
>
> Thanks for your insightful review. We address your questions and concerns below.
>
> **Q1. Lack of standard deviation or variance reporting, despite the paper's core focus on variance reduction.**
>
> Thank you for the suggestion. We conducted a large-scale empirical study to **directly quantify the reduction of coefficient of variation** (CV, a normalized measure of standard deviation) and show that our method reduces CV by **−16.9% to −36.8%** compared with DAPO Norm and Dr. GRPO Norm across different model types and parameter groups.
>
> Specifically, we conduct the empirical study following the methodology in Section 2.2:
>
> - We sample 1024 problems from Open Reasoner Zero.
> - For each problem, we sample 128 responses, producing **131,072 total responses**.
> - For each problem, we approximate the ground-truth gradient $E(g)$ by averaging the gradients of its 128 responses.
> - For each response, we compute its gradient deviation $||g - E(g)||^2$.
> - We measure this deviation on the Q proj, K proj, and first MLP projection in the last layer of **Qwen2.5-7B and Llama-3.2-3B-Instruct**.
> - Based on the gradient deviation, we can now quantify the CV reduction compared with DAPO Norm and Dr. GRPO Norm, across the entire 131K samples.
>
> The results are shown below:
>
> | Model               | Q Proj       | K Proj       | MLP 1st Proj  |
> |--------------------|--------------|--------------|---------------|
> | Llama-3.2-3B-Instruct | -36.8%       | -34.7%       | -31.4%        |
> | Qwen2.5-7B           | -22.8%       | -22.0%       | -16.9%        |
>
> These results strongly support our claim that **VL Norm achieves substantial variance reduction in practice**.
>
> **Q2. In Figure 6, GRPO Norm maintains similarly low entropy, yet the paper presents this as a unique property of VL Norm. Could the authors clarify whether VL Norm has distinct stability benefits beyond what GRPO already offers?**
>
> We do **not** claim VL Norm further reduces variance compared with GRPO.  The advantage of VL Norm over GRPO lies in its **unbiasedness**.
>
> Theoretically, as shown in Table 1, $E(g)$ under GRPO has a **length-dependent bias** and does not faithfully approximate the true policy loss. When the response length increases, GRPO produces smaller effective updates in gradient norm, which slows down convergence.
>
> Empirically, in the first two subfigures of Figure 2:
>
> - GRPO and VL Norm both maintain stable updates in the early stage.
> - GRPO's updates **saturate** later in training.
> - VL Norm continues to improve and achieves **higher final accuracy**.
>
> Thus, the advantage of VL Norm is correctness and unbiased updating, compared with GRPO.
>
> **Q3. VL Norm may exacerbate length bias, contrary to its motivation. In Figure 7, VL Norm exhibits a sharp increase in response length (especially in Math 7B), unlike the baselines.**
>
> We make two clarifications.
>
> First, VL Norm does **not** promote any length bias.  If we take the expectation of the aggregated gradient $g$, the sum of the introduced factors in VL Norm is exactly $1/M$, where $M$ is a constant. Therefore the approximation remains the true policy gradient, without introducing any preference for longer responses. This holds without additional assumptions. VL Norm only aggregates gradients without rewarding or penalizing long answers.
>
> Second, in the Math task, the length increase coincides with **higher test accuracy**, as shown in Figure 5. In other tasks such as CountDown, VL Norm also outperforms baselines, while the response length is similar or even shorter (Figure 8 and Figure 11). This suggests that the length change is a task-specific phenomenon rather than an inherent effect of VL Norm.
>
> Overall, the main purpose of VL Norm is to promote stable and correct RLVR training without promoting or discouraging long responses, and it is unbiased without additional assumptions.
>
> **Q4. In CountDown 7B (Table 2), GRPO Norm actually outperforms VL Norm. Why is this not discussed?**
>
> Thanks for pointing this out. We briefly mentioned this observation in Section 4.2, and add three justifications here:  (1) In this setting, VL Norm still clearly improves over the other three baselines;  (2) the score difference between VL Norm and GRPO Norm is small (0.821 vs. 0.827); and (3) VL Norm reaches this accuracy roughly 100 steps earlier than GRPO Norm.
>
> Besides, in 5 out of 6 settings VL Norm surpasses all four baselines, and in the remaining setting it outperforms three baselines and is close to the last one. We therefore conclude that VL Norm is a strong and promising aggregation method for RLVR.

---

> > ### Comment · Reviewer_Z4eq · 2025-11-25
> >
> > Thank you for the thorough and constructive review. I appreciate your feedback and will keep my score unchanged.

---

> > > ### Author Response · Authors · 2025-11-28
> > >
> > > Thank you for the acknowledgement. Please feel free to reach out with any further questions.

---

### Official Review · Reviewer_wEc6 · 2025-10-28

**Soundness:** 2
**Presentation:** 3
**Contribution:** 3
**Rating:** 6
**Confidence:** 4

**Summary:**

This paper introduces VL Norm (Variance-reduced Length-dependent Normalization), a new loss aggregation method designed to address challenges in Reinforcement Learning with Verifiable Rewards (RLVR), especially the variability in response lengths during training. RLVR has shown strong potential in improving the reasoning capabilities of large language models (LLMs), but it faces the challenge of gradient instability due to fluctuating response lengths.  The proposed VL Norm not only provides an unbiased estimate of the true policy loss but also minimizes gradient variance, achieving superior training stability and performance.

**Strengths:**

VL Norm introduces a fresh approach from statistical perspective to loss aggregation by addressing the length-dependent variability in RLVR tasks. The method's ability to provide both unbiased estimations and minimized variance is a valuable contribution to this field.

**Weaknesses:**

The paper heavily relies on the assumption that the variance structure of the token-level gradient grows proportionally with the response length. However, I found Figure 3 unclear. Specifically, how the Q, K, V matrices reflect the variability of the token-level gradient. Could you provide more details about the computational process involved in this and how these matrices accurately capture this variability?

**Questions:**

1. Could you further give an explanation about the connection and distinction between VL-norm and GSPO [1]. GSPO modifies the importance sampling weight
2. DAPO employs four key techniques that can significantly impact the variance of the gradient. For instance, the reward shaping strategy  avoid excessively long responses, which can significantly reduce the variance of the gradient. Does it seem fair to directly compare the variance across DAPO and other methods?
3. What is the impact of the hyperparameter $\alpha$ on the performance of VL Norm? Have the authors conducted experiments  to evaluate the sensitivity of VL Norm’s performance to variations in this hyperparameter?
4. The experiments mainly focus on CountDown and Math tasks. Could the authors conduct additional experiments on tasks with more complex reward structures, or tasks that require long-term decision-making?



[1] Group Sequence Policy Optimization. Qwen Team, Alibaba.

**Details Of Ethics Concerns:**

No ethical concerns are apparent in the paper.

---

> ### Author Response · Authors · 2025-11-19
>
> Thanks for your insightful review. We address your questions and concerns below.
>
> **Q1. Could you further give an explanation about the connection and distinction between VL-norm and GSPO [1].**
>
> To our understanding, **VL Norm is orthogonal to GSPO**.
>
> GSPO focuses on correcting the **importance sampling ratio** in RL training. In GRPO, importance sampling is applied at the **token level**, whereas GSPO argues that it should instead be applied at the **sequence level**. GSPO places the importance sampling ratio on the whole sequence and discards samples whose off-policyness exceeds a certain threshold.
>
> However, this still does not address the variance issue caused by long responses. Even among the unclipped samples, longer sequences still exhibit **larger gradient variance**. VL Norm directly addresses this problem and can be **combined with GSPO** to further stabilize RL training.
>
> **Q2.  Does it seem fair to directly compare the variance across DAPO and other methods?**
>
> Our main research goal is to study how different **normalization methods** influence RLVR. To control for confounding factors, we adjust **only** the normalization method and evaluate the effects in the main experiments. For clarity, we always refer to the method as **"DAPO Norm"**, not "DAPO".
>
> We also note that DAPO introduces additional techniques such as overlong filtering and soft punishment to control response length. In Section 4.3, we provide comparisons with these techniques and find that **VL Norm significantly outperforms them**.
>
> **Q3. What is the impact of the hyperparameter on the performance of VL Norm?**
>
> We report results for $\alpha \in \{0.5, 0.75, 1.0\}$ in Table 3.  Across all $\alpha$-s, the **average performance over the four experimental settings** consistently exceeds all baseline methods.
>
> The best average performance is achieved with $\alpha = 1.0$, which aligns with our theoretical analysis showing that the gradient variance is minimized at this value.
>
> **Q4. The experiments mainly focus on CountDown and Math tasks. Could the authors conduct additional experiments on tasks with more complex reward structures, or tasks that require long-term decision-making?**
>
> Thank you for the suggestion. Recent agentic tasks, such as multi-turn coding agents and deep research agents, can produce rollouts with **over one hundred turns** [1].  Such long-horizon settings naturally lead to **even larger gradient variance**, further motivating the need for VL Norm to provide unbiased, variance-minimized updates. We plan to include evaluations on these tasks in future experiments.
>
> [1] Beyond Ten Turns: Unlocking Long-Horizon Agentic Search with Large-Scale Asynchronous RL

---

> > ### Comment · Reviewer_wEc6 · 2025-11-26
> >
> > Thank you for the feedback! I will keep my initial score.

---

> > > ### Author Response · Authors · 2025-11-28
> > >
> > > Thank you for the acknowledgement. Please feel free to reach out with any further questions.

---

### Official Review · Reviewer_iNhA · 2025-10-29

**Soundness:** 2
**Presentation:** 3
**Contribution:** 3
**Rating:** 4
**Confidence:** 4

**Summary:**

This paper identifies that existing RLVR methods suffer from biased estimation or high variance when performing response length normalization. The authors innovatively propose the VL Norm method, framing length normalization as a minimum variance unbiased estimation problem, which stabilizes RLVR training and facilitates convergence.

**Strengths:**

1. Although the length normalization issue addressed in the paper is a minor component of RLVR algorithms, it significantly impacts training stability and convergence speed. The authors provide a novel perspective by analyzing gradient variance.
2. The theoretical analysis of minimum variance unbiased estimation is highly convincing. Table 1 clearly demonstrates the superiority of the proposed method through comparisons of gradient mean and gradient variance with baselines.
3. The experimental results in Section 4 are analyzed in great detail.

**Weaknesses:**

1. All experiments are conducted on Qwen models. As noted in [1][2], the Qwen series may suffer from data contamination issues in evaluation datasets such as MATH500. Therefore, I strongly recommend that the authors perform additional experiments on more open-source models (e.g., the Llama3 series) to enhance the robustness of their conclusions. I would be willing to raise my score if more experimental results are provided.
2. The entire theoretical framework is based on the assumption that gradient variance is positively correlated with response length. However, the authors only use statistics from one or two samples to support this claim, which is insufficient. Theoretical support for this argument would be preferable. Alternatively, the authors should conduct experiments with more samples, model series, and layer types.
3. The proposed method introduces a hyperparameter alpha, but the final experimental results (Table 4) do not show a convincing improvement.

**Reference**

[1] Wu et al., Reasoning or Memorization? Unreliable Results of Reinforcement Learning Due to Data Contamination, arxiv 2025

[2] Shaoe et al., Spurious Rewards: Rethinking Training Signals in RLVR, arxiv 2025

**Questions:**

1. Why do the main experiments in Fig. 2 and Fig. 5 not show the complete results of VL Norm? Is this due to the instability of RL training causing performance degradation in later stages?
2. [1] also bases on the assumption that gradient variance is positively correlated with response length and starts from the goal of minimizing gradient variance. Could the author briefly explain the similarities and differences between this paper and your paper? If possible, could this method be included as a baseline in your paper?

Reference

[1] Hao et al., On-Policy RL with Optimal Reward Baseline, arxiv 2025

---

> ### Author Response · Authors · 2025-11-19
> **Response to Reviewer iNhA (1/2)**
>
> Thanks for your insightful review. We address your questions and concerns below.
>
> **Q1.  I strongly recommend that the authors perform additional experiments on more open-source models (e.g., the Llama3 series) to enhance the robustness of their conclusions.**
>
> Sure. We further evaluate our method on the **Llama-3.2-3B-Instruct** model using both the CountDown and Math tasks. **The paper is updated with the full training dynamics, including testing scores (evaluated every 50 steps), entropy, and response length, in Appendix A.6.** Llama-3.2-3B-Instruct model is trained for 500 steps with a maximum response length of 3072, and we set $\alpha=1$ for the proposed method VL Norm. All other training configurations follow those used for the Qwen models.
>
> Across both tasks, VL Norm consistently outperforms the baseline methods (GRPO Norm, DAPO Norm, and Dr. GRPO Norm) in the best test score, as shown in following table. On CountDown, our method achieves higher test scores at every evaluation point from step 100 to step 500. For Math, while the improvements are more modest, VL Norm remains the best-performing method from step 350 onward. The complete curves are provided in Figures 11–12 in the revised paper.
>
>
> | Method        | Llama; CountDown AVG@8 | LLama; Math W.AVG@8 |
> |---------------|------------------|----------------|
> | GRPO Norm     | 0.750            | 0.400          |
> | DAPO Norm     | 0.720            | 0.405          |
> | Dr. GRPO Norm | 0.722            | 0.402          |
> | VL Norm (Ours)      | **0.776**            | **0.410**          |
>
>
> We also make several observations regarding entropy and response length. In CountDown, VL Norm and GRPO Norm maintain relatively stable entropy, whereas DAPO Norm and Dr. GRPO Norm fluctuate more significantly, which is consistent with our experiments on Qwen models. In Math, the entropy changes are similar across methods, but VL Norm produces slightly longer responses in the middle stage of training, which may be related to its stronger performance.
>
> **Q2. The entire theoretical framework is based on the assumption that gradient variance is positively correlated with response length. However, the authors only use statistics from one or two samples to support this claim, which is insufficient.**
>
> To further validate this main assumption, we conduct a large-scale empirical study following the methodology in Section 2.2:
>
> - We sample 1024 problems from Open Reasoner Zero.
> - For each problem, we sample 128 responses, producing **131,072 total responses**.
> - For each problem, we approximate the ground-truth gradient $E(g)$ by averaging the gradients of its 128 responses.
> - For each response, we compute its gradient deviation $||g - E(g)||^2$.
> - We measure this deviation on the Q proj, K proj, and first MLP projection in the last layer of **Qwen2.5-7B and Llama-3.2-3B-Instruct**.
> - We then compute the **Pearson correlation** between response length and gradient deviation across the entire 131K samples.
>
> The results are shown below:
>
> | Model                  | Q Proj | K Proj | MLP 1st Proj |
> |-----------------------|--------|--------|---------------|
> | Llama-3.2-3B-Instruct | 0.788  | 0.800  | 0.907         |
> | Qwen2.5-7B            | 0.740  | 0.714  | 0.683         |
>
> Across models and parameter types, the Pearson R ranges from 0.68 to 0.91, indicating **strong positive correlation** between response length and gradient deviation. A linear model predicting deviation from length yields an $r^2$ between 0.5–0.8, demonstrating a **medium to strong linear relationship**. These large-scale empirical findings strongly support our assumption that **the variance of the unnormalized gradient grows linearly with response length**. We additionally refer you to Figure 3 in the main text and Figures 9–10 in the appendix for visualizations of this trend, of different samples and parameter types.

---

> ### Author Response · Authors · 2025-11-19
> **Response to Reviewer iNhA (2/2)**
>
> **Q3. Why do the main experiments in Fig. 2 and Fig. 5 not show the complete results of VL Norm? Is this due to the instability of RL training causing performance degradation in later stages?**
>
> We now provide the full training curves for all six settings in Appendix A.7 of the updated submission. While the performance typically saturates after the best checkpoint, it does **not** exhibit severe degradation. In most cases, the proposed VL Norm is better than baselines after the best checkpoint.
>
> We also report the *monotonicity score*, computed as the Pearson correlation between the training step and the test metric **over the entire training course**. As shown below, VL Norm achieves the highest monotonicity score (0.977), indicating that it encourages more stable and steadily improving training dynamics. Please refer to Table 5 for detailed numbers.
>
> | GRPO Norm | DAPO Norm | Dr. GRPO Norm | Dr. GRPO | VL Norm (Ours) |
> |-----------|-----------|----------------|----------|-------|
> | 0.779     | 0.848     | 0.719          | 0.729    | **0.977** |
>
>
> **Q4. The proposed method introduces a hyperparameter alpha, but the final experimental results (Table 4) do not show a convincing improvement.**
>
> We report results for $\alpha \in {0.5, 0.75, 1.0}$ in Table 3. Across all three choices, the average performance over the four experimental settings consistently surpasses all baseline methods. Moreover, the best average result is obtained with $\alpha = 1.0$, which aligns with the theoretical analysis showing that the gradient variance is minimized at this point. These results support our claim that VL Norm offers a more effective normalization strategy for RLVR.
>
> **Q5. Similarities and comparison with OPO.**
>
> Both our method and OPO aim to minimize the gradient variance, but they optimize **different variables**.
>
> - OPO adjusts only the **baseline**, keeping the aggregation factors unchanged.
> - Our method optimizes the **normalization factors**, which directly scale the unnormalized gradient.
>
> This leads to different impacts. When the response is long and the gradient variance grows with length, dividing by a large normalization factor $N$ reduces variance by about $1/N$. In contrast, modifying the baseline provides much smaller changes, especially in RLVR where the baseline typically lies between 0 and 1. The baseline optimized by OPO still stays close to the natural baseline computed as a simple average.
>
> Empirically, we evaluate OPO using the Qwen2.5-7B model on the Math task, following the paper's configuration (OPO on top of GRPO normalization). OPO brings only a small improvement over GRPO (+0.004 in W.AVG), and remains far behind our method.
>
> | Method     | MATH500 | Minerva | AMC      | AIME2024 | AVG     | W.AVG   |
> |------------|---------|---------|----------|----------|---------|---------|
> | GRPO   | 0.777   | 0.275   | **0.488** | 0.117    | 0.414   | 0.573   |
> | OPO        | 0.776   | 0.300   | 0.446    | 0.150    | 0.418   | 0.577   |
> | Ours   | **0.794** | **0.302** | 0.483    | **0.167** | **0.436** | **0.592** |

---

> ### Comment · Reviewer_iNhA · 2025-11-27
>
> Thank you for your detailed feedback. My concerns have been fully addressed, and I will be raising my score accordingly.

---

> > ### Author Response · Authors · 2025-11-27
> >
> > We sincerely thank Reviewer iNhA’s acknowledgement of our rebuttal and for the increased rating.

---

### Official Review · Reviewer_cu3P · 2025-11-01

**Soundness:** 3
**Presentation:** 3
**Contribution:** 3
**Rating:** 6
**Confidence:** 3

**Summary:**

This paper investigates the bias and variance of gradients induced by response length, along with related normalization methods. The authors analyze how varying response lengths affect policy loss and propose a normalization technique that achieves an unbiased gradient estimate with minimal variance. Extensive experiments are conducted to demonstrate that the proposed method improves both performance and training stability (in terms of entropy dynamics).

**Strengths:**

- The paper is well written and easy to follow.

- The authors present a novel and well-founded perspective on length normalization, which is both important and closely related to the training stability of RL algorithms. They provide a theoretical analysis of the effects of different length normalization methods and propose a new normalization approach based on this theoretical insight.

- The experimental results are sound, evaluating both the final performance and the training stability of the proposed method.

**Weaknesses:**

- The main assumption (even in the derivation of VL Norm) is the independence of token gradients (i.e., zero covariance; Line 174). However, intuitively, sequential tokens are unlikely to be independent. Although Figure 4 shows that gradient variance is correlated with response length, I am still wondering how this independence assumption might affect the results. Could the authors provide more discussion or justification regarding this assumption?
- The experimental datasets seem relatively “easy” compared to the AIME dataset commonly used in RLVR papers. The length issue would likely be more significant on such a challenging dataset, as it requires the model to generate longer responses to solve the problems. In other words, evaluating on a more difficult dataset like AIME would better highlight the importance of the length normalization issue.

**Questions:**

Please refer to Weaknesses part.

---

> ### Author Response · Authors · 2025-11-19
>
> Thanks for your insightful review. We address your questions and concerns below.
>
> **Q1. The main assumption is the independence of token gradients. However, intuitively, sequential tokens are unlikely to be independent. Although Figure 4 shows that gradient variance is correlated with response length, I am still wondering how this independence assumption might affect the results. Could the authors provide more discussion or justification regarding this assumption?**
>
> Our core assumption is **that the variance of the unnormalized gradient grows approximately linearly with the response length**. Intuitively, the unnormalized gradient is a sum over N token-level gradients. Each token contributes randomness, and summing N such terms naturally leads to a variance that increases roughly linearly with N. In the theoretical analysis, we assume token-level independence for analytical convenience. But it is not essential for our method. The method only relies on the assumption that **gradient variance increases approximately linearly with response length**, rather than requiring strict independence between token-level gradients.
>
> To further validate this main assumption, we conduct a large-scale empirical study following the methodology in Section 2.2:
>
> - We sample 1024 problems from Open Reasoner Zero.
> - For each problem, we sample 128 responses, producing **131,072 total responses**.
> - For each problem, we approximate the ground-truth gradient $E(g)$ by averaging the gradients of its 128 responses.
> - For each response, we compute its gradient deviation $||g - E(g)||^2$.
> - We measure this deviation on the Q proj, K proj, and first MLP projection in the last layer of Qwen2.5-7B and Llama-3.2-3B-Instruct.
> - We then compute the **Pearson correlation** between response length and gradient deviation across the entire 131K samples.
>
> The results are shown below:
>
> | Model                  | Q Proj | K Proj | MLP 1st Proj |
> |-----------------------|--------|--------|---------------|
> | Llama-3.2-3B-Instruct | 0.788  | 0.800  | 0.907         |
> | Qwen2.5-7B            | 0.740  | 0.714  | 0.683         |
>
> Across models and parameter types, the Pearson R ranges from 0.68 to 0.91, indicating **strong positive correlation** between response length and gradient deviation. A linear model predicting deviation from length yields an $r^2$ between 0.5–0.8, demonstrating a **medium to strong linear relationship**. These large-scale empirical findings strongly support our assumption that **the variance of the unnormalized gradient grows linearly with response length**. We additionally refer you to Figure 3 in the main text and Figures 9–10 in the appendix for visualizations of this trend.
>
> To summarize, our method does **not** depend on token-level independence. What matters is the empirically verified **linear growth of gradient variance as response length increases**, which is consistently observed across models and parameter types.
>
> **Q2. The experimental datasets seem relatively "easy" compared to the AIME dataset commonly used in RLVR papers. The length issue would likely be more significant on such a challenging dataset, as it requires the model to generate longer responses to solve the problems.**
>
> Thank you for the suggestion. We have conducted experiments on the AIME2024 dataset, and the detailed results are presented in Table 4. On both the 3B model with a maximum response length of 8192 and the 7B model, VL Norm achieves the best performance among all baselines. On the 3B model with a 3072 response length, it obtains the second-best performance.
>
> We plan to further extend our evaluation to more recent open-source models, such as Qwen3-4B, and assess performance on both AIME2024 and AIME2025.
>
> Besides, recent agentic tasks, such as multi-turn coding agents and deep research agents, can produce rollouts with **over one hundred turns** [1].  Such long-horizon settings naturally lead to **even larger gradient variance**, further motivating the need for VL Norm to provide unbiased, variance-minimized updates. We plan to include evaluations on these tasks in future experiments.
>
> [1] Beyond Ten Turns: Unlocking Long-Horizon Agentic Search with Large-Scale Asynchronous RL

---

### Author Response · Authors · 2025-11-28
**General Remarks on the Rebuttal**

Dear Area Chair and Reviewers,

**We sincerely thank all of you for your effort with our work.**

The reviewers' feedback substantially strengthened the paper. During the rebuttal, we responded to all raised concerns thoroughly. Reviewer `iNhA` has updated his/her score from 4 to 6 after we fully addressed the raised concerns, including **conducting additional experiments on the Llama model and providing stronger empirical validation for the gradient–variance assumption.**

Below, we present a clear summary of the paper's strengths, the main concerns raised across reviews, and the  revisions we have made for your reference.

> ## I. Paper Strengths

*(We summarize each strength in one sentence and directly quote reviewers’ original words to accurately reflect their assessments.)*

First, the reviewers most appreciated the **novelty of our method and the theoretical soundness of our modeling**: VL Norm was recognized as offering “a novel and well-founded perspective on length normalization” (Reviewer `cu3P`), providing “highly convincing theoretical analysis of minimum-variance unbiased estimation” (Reviewer `iNhA`), introducing “a fresh approach from a statistical perspective” (Reviewer `wEc6`), and presenting an “elegant MVUE formulation that generalizes prior methods” (Reviewer `Z4eq`).

Second, the reviewers highlighted the **breadth and depth of our empirical coverage**: experiments were described as “sound and evaluate both final performance and training stability” (Reviewer `cu3P`), the empirical analysis was said to be “conducted in great detail” (Reviewer `iNhA`), and our study was praised for “comprehensive experiments across multiple tasks, model sizes, and maximum response lengths” (Reviewer `Z4eq`).

Finally, the reviewers emphasized the **practical relevance** of our contribution: the work addresses a component that, while small, “significantly impacts training stability and convergence speed” (Reviewer `iNhA`) and tackles an issue “important and closely related to the training stability of RL algorithms” (Reviewer `cu3P`), underscoring VL Norm’s practical value for RLVR training.

> ## II. Common Weaknesses and Our Revisions

Reviewers raised common concerns about the **validation of our key assumption** that gradient variance increases approximately linearly with response length. This is explicitly raised by Reviewer `cu3P`, `iNhA`, and `wEc6`. Reviewer `Z4eq` also emphasized the need for stronger empirical support for this central assumption and the variance reduction.

In response to this concern, we conducted a large-scale empirical study using **131,072 responses** across **two different model families**, showing a strong positive correlation between response length and gradient deviation. A simple linear predictor explains the variance with **$r^2$ between 0.5–0.8**, demonstrating a medium-to-strong linear relationship. These results significantly strengthen the assumption and are now included in **Appendix A.8** of the updated manuscript.

We also respond to different reviewers' specific concerns:

- **Reviewer `cu3P`**: We clarified that we **did run experiments on the AIME dataset** and report the results.
- **Reviewer `iNhA`**: We added **new experiments on Llama models** to address model diversity and potential contamination issues in **Appendix A.6**, and we included **full training curves** in **Appendix A.7**.
- **Reviewer `wEc6`**: We expanded discussion on connections to **GSPO**, **DAPO**, and analyzed the influence of hyperparameters more clearly.
- **Reviewer `Z4eq`**: We clarified the **unbiasedness of VL Norm**, its advantages over GRPO, and addressed concerns regarding stability.

We believe these revisions and supplementary results thoroughly address the reviewers' concerns.  Thank you again, and we are available for any further clarification if needed.

Sincerely,

**Authors of Submission 10262**

---

### Meta-Review · Area_Chair_eCw2 · 2026-01-03

**Summary:**

The initial scores were 6, 6, 4, and 4. I appreciate the authors' hard work during the rebuttal period. However, all reviewers remain borderline, and although the authors provided additional results and clarifications, the theoretical concerns were not fully resolved. Moreover, no statistical significance tests were provided in the experiments. This work looks promising and could be impactful, but in its current form it falls short of the bar for acceptance at ICLR. I encourage a revised resubmission.

**Reviewer Concerns:**

A lot of the initial concerns were addressed. The largest addition is the empirical validation that corresponds to the assumption of length and variance, but the rebuttal did not fully address the theoretical side. Coefficient of variation reductions were added, but statistical significance tests or conf interval with multiple seeds, requested by reviewer Z4eq, remained missing.

**Reviewer Scores:**

One reviewer with rating 4 mentioned they will maintain their score. Potentially reviewer(s) could have increased their score based on the added results, if we had a full discussion period, for example, finalized with 6664.

---

### Decision · Program_Chairs · 2026-01-26

Reject